# The potential of UAV-borne spectral and textural information for predicting aboveground biomass and N fixation in legume-grass mixtures

Esther Grüner *, Michael Wachendorf, Thomas Astor

Grassland Science and Renewable Plant Resources, Organic Agricultural Sciences, Universität Kassel, Witzenhausen, Germany

* gnr@uni-kassel.de, esther.gruener@uni-kassel.de

## Abstract

Organic farmers, who rely on legumes as an external nitrogen (N) source, need a fast and easy on-the-go measurement technique to determine harvestable biomass and the amount of fixed N ($N_{Fix}$) for numerous farm management decisions. Especially clover- and lucerne-grass mixtures play an important role in the organic crop rotation under temperate European climate conditions. Multispectral sensors mounted on unmanned aerial vehicles (UAVs) are new promising tools for a non-destructive assessment of crop and grassland traits on large and remote areas. One disadvantage of multispectral information and derived vegetations indices is, that both ignore spatial relationships of pixels to each other in the image. This gap can be filled by texture features from a grey level co-occurrence matrix. The aim of this multi-temporal field study was to provide aboveground biomass and $N_{Fix}$ estimation models for two legume-grass mixtures through a whole vegetation period based on UAV multispectral information. The prediction models covered different proportions of legumes (0–100% legumes) to represent the variable conditions in practical farming. Furthermore, the study compared prediction models with and without the inclusion of texture features. As multispectral data usually suffers from multicollinearity, two machine learning algorithms, Partial Least Square and Random Forest (RF) regression, were used. The results showed, that biomass prediction accuracy for the whole dataset as well as for crop-specific models were substantially improved by the inclusion of texture features. The best model was generated for the whole dataset by RF with an rRMSE of 10%. For $N_{Fix}$ prediction accuracy of the best model was based on RF including texture (rRMSEP = 18%), which was not consistent with crop specific models.

## Introduction

The production of industrial fertilizer containing nitrogen (N) as an essential element of plant growth increased global food production in the last decades. Nevertheless, the excessive use of

**Data Availability Statement:** The data set, which is necessary to replicate our study findings is at PANGAEA via gfbio: https://doi.org/10.1594/PANGAEA.914667.

 

**Funding:** The author(s) received no specific funding for this work.

**Competing interests:** The authors have declared that no competing interests exist.

N in agriculture caused different environmental problems, like e.g. eutrophication and pollution of water [1] and is also linked to air pollution and climate change [2]. To reduce these negative impacts on the environment, organic agriculture uses a reduced N supply as it follows the idea of a closed nutrient cycle and, thus, is considered as a sustainable and low-input agricultural system [3]. As mineral N fertilizer is prohibited in organic agriculture, N is the primary limiting nutrient for many farms [4]. Beside organic fertilizer, organic farmers rely on legumes as an external N supplier, due to the ability of fixing atmospheric N in symbiosis with *Rhizobium* bacteria [5]. Especially, legume-grass mixtures, like e.g. clover-grass or lucerne-grass, cultivated for 1–3 years and cut 3–5 times a year, are an inherent part of the organic crop rotation in temperate European climates. These mixtures detain even higher amounts of N than the same solely cultivated legume [6] and are used as green manure or forage for livestock as well as for biogas plants. Total annual N fixation varies strongly and can reach up to over 300–500 kg N ha$^{-1}$ year$^{-1}$ depending on the legume species [7–9]. Beside N acquisition, legume-grass mixtures provide additional positive effects on the subsequent cash crop, i.e. enhanced product quality and soil fertility as well as weed suppression [10–13]. For an efficient nutrient management on farm level both with and without livestock, knowledge on the actual amount of biomass harvested as well as on the total annual biomass is necessary [14]. Furthermore, the amount of fixed N ($N_{Fix}$) is a valuable information, as it contributes to the N cycle of the farm. In order to understand the dynamics of legume-grass mixtures over the entire growing season, multi-temporal studies are required. It is well known that legume proportion varies strongly in practical farming which substantially affects total annual $N_{Fix}$ [15].

Traditional methods and devices to estimate biomass in practical grassland management, like destructive biomass sampling, ruler height or rising plate meter measurements are commonly used but are labour and time consuming. Methods to estimate $N_{Fix}$ are very expensive and need a lot of laboratory work, like isotope measurement of the $^{15}N/^{14}N$ ratio of the natural abundance or by the isotope dilution method [16]. Another method is the nitrogen difference method, where the N content in the aboveground biomass of a N-fixing plant and a non-N-fixing plant as reference is analyzed and the difference is calculated [17]. Høgh-Jensen et al. [18] generated a model for legume-grass mixtures, which includes additional sinks of fixed N, like the amount of fixed N in the below-ground biomass of the fixing plant, as well as of the grass. Nevertheless, farmers need efficient on-the-go measurement techniques for biomass and $N_{Fix}$ estimation for short-term management decisions.

Solutions for new non-destructive measurement techniques, covering large areas in short time, can be found in the field of remote sensing, which developed rapidly in the last decades in agricultural and grassland science. An overview about different applications of remote sensing in grasslands is given in Schellberg et al. [19] and Wachendorf et al. [20]. A very promising platform for sensors are unmanned aerial vehicles (UAVs), which are becoming increasingly important for agricultural crops [21,22]. There exists a variety of different optical sensors, to be mounted on UAVs for the assessment of agricultural crops. These sensors capture the reflecting light of vegetation, ranging from low-cost costumer-grade cameras for RGB (red, green, blue) images to multi- and hyperspectral sensors, which cover also non-visible spectral bands [23–25].

To compensate background noise from soil or atmosphere in the spectral reflection and to increase the sensitivity for vegetation traits, vegetation indices (VIs) can be calculated [26]. The most known VI, which uses the red and near infrared (NIR) bands, is the Normalized Difference Vegetation Index (NDVI) [27]. Nevertheless, NDVI has its limitations, as it saturates at high biomass and LAI values [23,28] and is varying with soil colour [29] and due to atmospheric effects [30]. In the recent decades more than 100 VIs were developed, which vary in calculation and proportion of different spectral bands for specific applications and utilization

 

for biophysical and -chemical features of vegetation [31]. For estimating different vegetation features in grassland, VIs are already well-studied tools like for biomass yield [32,33], leaf area index (LAI) [34] and quality [35].

One aspect with the common use of spectral bands and VIs is that they ignore spatial variability of the grey level values (pixels) within the neighbourhood of each pixel. This gap can be covered by texture analysis, which is more complex to quantify than spectral information [36]. Texture is the spatial variation within an image, which correlates with the structure and heterogeneity of vegetation [37]. Furthermore, Culbert et al. [38] documented that texture features can vary depending on the observed vegetation and its phenological stage, which makes it an interesting tool for multi-temporal studies. These facts are gaining more attention in research studies, mainly focusing on land cover classification [39–41], vegetation modelling [42,43] and structure [44,45] as well as forest biomass estimation [46,47]. For agricultural crops, there exist, to our best knowledge, only two studies using texture features in combination with UAV multispectral information. In a multi-temporal study over two years, Zheng et al. [48] and Li et al. [49] used a multispectral sensor on a UAV to estimate rice biomass and LAI including different cultivars, varying seed densities and N levels. The authors concluded that the combination of spectral and texture information is a promising method for biomass estimation. Few studies have explored the effectiveness of texture features in permanent grass- and rangeland and mainly focused on the classification of vegetation [50]. Gebhardt and Kühbauch [51] as well as van Evert et al. [52] found a high detection accuracy for *Rumex obtusifolius* in grasslands by using texture features. Guo et al. [53] compared two grassland management systems (i.e. grazed and non-grazed) and successfully described differences in spatial heterogeneity of these grasslands. The most popular method of gaining texture information of remotely sensed images follows a statistical approach, named Grey Level Co-occurrence Matrix (GLCM), which can be used to calculate so-called second order texture features [54]. While first-order texture features are calculated directly from occurring grey level values (e. g. variance) within a certain area (window), second-order texture features consider the spatial relationship between these pixels, also called co-occurrence. Therefore, these features are more computationally intensive, but have a greater potential to represent the structure of vegetation. Nevertheless, there exist no studies which estimate biomass in terms of aboveground biomass and $N_{Fix}$ of legume-grass mixtures using texture features in combination with multispectral information.

The aim of this study is to develop harvestable biomass and aboveground $N_{Fix}$ estimation models from UAV multispectral imaging of legume-grass mixtures with varying legume proportions (0–100%). To deal with multispectral data, which usually suffers from multicollinearity, machine learning algorithms can be a solution, especially when the number of predictors is higher than the number of samples in the dataset. The specific objectives of this study were: (1) to develop harvestable biomass and aboveground $N_{Fix}$ prediction models for mixtures with two different legumes by machine learning algorithms for a whole growing season; (2) to compare the prediction accuracy of these models with and without the inclusion of texture features; (3) to identify key variables of the resulting models; (4) to compare the measured and predicted total annual dry matter biomass and $N_{Fix}$ including all cuts in one growing season.

## Material and methods

### Experimental site and ground truth data

The field experiment was conducted in 2017 and 2018 in Neu-Eichenberg at the experimental farm of the Universität Kassel, which is located in northern Hesse, Germany (51°23' N, 9°54' E, 227 m above sea level). The soil is characterized as a silty clay loam and oil radish was

cultivated as a preceding crop. As the experimental farm is managed organically, no pesticides or mineral N fertilizer were applied. Long-term average annual precipitation at the site is 687 mm, but the amount of rainfall in 2018 was unusually low (350 mm) and led to severe drought throughout the whole vegetation period.

Field plots were established in autumn 2017 with a size of 1.50 m × 12 m and sown with a total seed rate of 35 kg ha$^{-1}$. The treatments consisted of two legume-grass-mixtures, clover- (CG) and lucerne-grass (LG), and additionally pure stands of legumes ($L_{CG}$, $L_{LG}$) and grass ($G_{CG}$, $G_{LG}$) of both mixtures. These six treatments were sown in four randomized replicates, resulting in 24 plots in total (Fig 1). CG contained 60% *Lolium multiflorum*, 30% *Trifolium pratense*, 5% *Trifolium hybridum* L. and 5% *Trifolium repens* L., whereas LG included 40% *Medicago sativa*, 20% *Festuca pratensis* Huds., 15% *Lolium perenne* L., 10% *Lolium multiflorum*, 10% *Trifolium pratense* and 5% *Phleum pratense* L (S1 Table).

In total eight biomass datasets for fresh (FM) and dry matter (DM) were obtained from three main harvests (17.05.2018, 20.06.2018, 03.08.2018) and to cover the whole vegetation period from five additional sub-sampling dates, whereas $N_{Fix}$ was determined only at main harvests. At each main harvest two aboveground biomass samples were collected in each plot on an area of 50 cm × 50 cm and cut at a stubble height of 5 cm. Biomass samples were weighed, dried for 48 h at 105°C and weighed again to determine DM. The remaining above-ground biomass was removed with a Haldrup forage plot harvester. Two additional samples of fresh biomass were taken. One sample of all plots was dried at 60°C for a later N content analysis. The second sample was taken only from the mixtures to determine the different fractions: legumes, grass, herbs and senesced material. The five sub-samples were taken between the main harvests at an area of 25 cm × 25 cm, restricted to the first 1.5 m of every plot to leave the remaining area undisturbed for the main harvests (Fig 1B).

N concentration in the biomass was assessed by an elemental microanalyzer (Elementar vario MAX CHN, Langenselbold, Germany) and N content in the aboveground biomass was determined by multiplication of N concentration and DM biomass. The difference method according to Stülpnagel [17] was used to quantify $N_{Fix}$ of the legumes (Eq 1):

$$N_{Fix} = N_L - N_R \tag{1}$$

where the N content of the mixtures and the pure stand of legumes were defined as the N fix-ing crop ($N_L$) and the pure stands of grass were used as the non-fixing reference crop ($N_R$).

## UAV image acquisition and data pre-processing

UAV flight missions were conducted before each cut (eight flights in total) in the morning at a nearly equal sun position. A low-cost quadrocopter (DJI Phantom 3 Advanced, Shenzhen, China) was used equipped with a multispectral sensor (Parrot Sequoia, MicaSense Inc, Seattle, USA). The sensor captured the reflected light in four separate bands: green (530–570 nm), red (640–680 nm), red edge (730–740 nm) and near-infrared (NIR; 770–810 nm) with a spatial resolution of 1.2 Megapixel (MP) as well as red, green, blue (RGB) images with a spatial resolu-tion of 16 MP. The sensor was supplied with an additional sunshine sensor, which was mounted on the top of the drone to capture the at-the-sensor irradiance for automatic calibra-tion of every picture. This radiometric calibration is done to eliminate variation in sunlight conditions during flight for the subsequent analysis. For two cuts the drone was flown at a flight altitude of 50 m above ground, where image overlap was 100%. Therefore, as a compro-mise between flight height and time, remaining cuts were flown at 20 m. All flight missions were flown manually, as due to the removal of the original camera automatic flight missions created major internal technical problems. The UAV was steered in a grid pattern through the

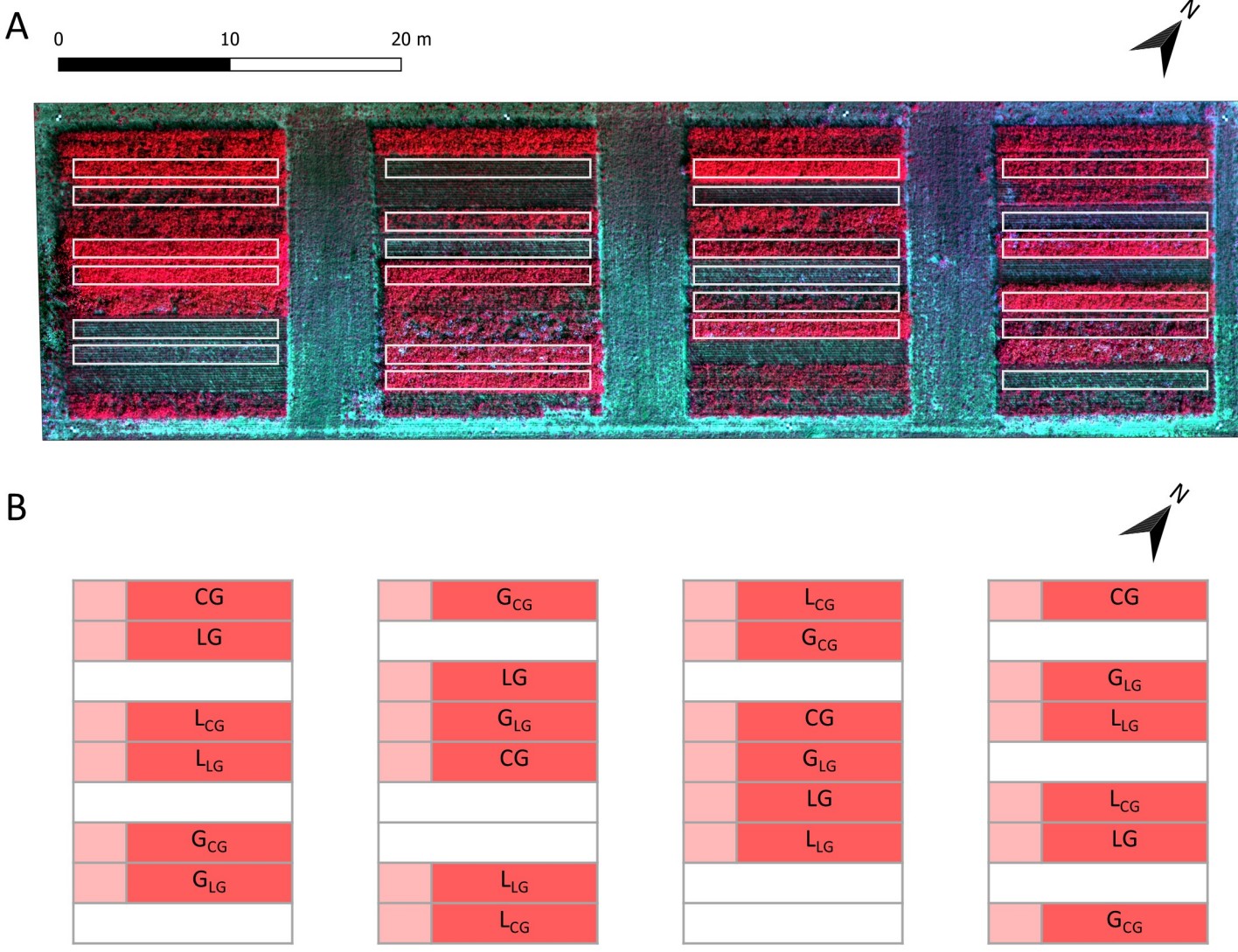

**Fig 1. Overview of experimental field and treatments.** (A) Orthomosaic of the experimental field showing the different plots (white border) one day before the third main harvest (02.08.2018) in false colours (NIR, red, green); (B) Design of experimental layout with treatments in four replicates: clover- (CG) and lucerne-grass (LG) and pure stands of legumes ($L_{CG}$, $L_{LG}$) and grasses ($G_{CG}$, $G_{LG}$) of the mixtures. Sampling area for main harvests (red) and the first 1.5 m for sub-sampling (light red).

experimental field at a speed of less than 1 m/s and the time trigger of the sensor was set to one image per 1.5 sec, resulting in about 1000 to 2000 images. Eight portable ground control points (GCPs; 25 cm × 25 cm) were evenly distributed in the pathways next to the plots and georeferenced by a differential global positioning system (DGPS, Leica, Germany) with a mean horizontal and vertical error of 0.02 m (S3 Table). Additionally, the plot corners were georeferenced once to delimit the individual plots.

Photogrammetric processing was done by means of structure from motion (SfM) with Agisoft PhotoScan Professional (Agisoft LLC, St. Petersburg, Russia) to generate multispectral orthomosaics for every flight mission. The overlapping images, including more than nine images for the area of interest, of each imported dataset were aligned with internal algorithms and image fitting techniques of the software. The position and orientation of each image was adjusted, and a sparse point cloud was created with the accuracy setting "high", a key point

limit of 40,000 and tie point limit of 1,000. Coordinates of the GCPs had to be placed in five images manually and automatic camera calibration was used to improve the accuracy of the sparse point cloud. In the next step a dense point cloud was created with "high" quality settings and a "mild" depth filtering to achieve a detailed image. The resulting output was a multispectral orthomosaic including the reflectance values of each band. Due to different flight altitude (50 and 20 m), ground resolution ranged between 2–4 cm. Therefore, orthomosaics were exported as a TIFF file with 4.5 cm resolution for unified conditions for subsequent analysis.

## Data analysis and machine learning

Extraction of reflectance information of the four bands and calculation of the texture features was achieved by Quantum Geographical Information System (QGIS 2.18.14, QGIS Development Team, Raleigh, NC, USA). Coordinates of the plot corners were used to create polygons, avoiding the first 1.5 m which were disturbed by biomass sub-sampling (Fig 1B). For the remaining area of 15.75 m$^2$ zonal statistic was applied for polygons of each plot and band to generate average values of every variable. An overview of the workflow including the variables and analysis is given in Fig 2.

**Texture features of images.**   Image texture can be described as the spectral and spatial variability of grey level values of an image. Haralick suggested a GLCM using 14 second order textural features of remotely sensed images [54]. As there exists no evidence in literature which features are best suitable for biomass prediction in grassland or legume-grass mixtures, eight of these GLCM texture features were used (Table 1), which were provided by the processing tool HaralickTextureExtraction of the Orfeo Toolbox library (OTB, open source, [55]) in QGIS. The eight Haralick texture parameters were computed for all four spectral bands separately with settings on default (window size: 2 × 2) and a texture set selection on "simple". The radiometric resolution was set to 16 bit.

**Vegetation index calculation.**   The spectral information of the orthomosaics was used to calculate a set of common VIs. Thirteen VIs using visible and red edge as well as NIR reflectance were selected (S2 Table), which were reported in literature for structural or biochemical characteristics of vegetation and grasslands [31,35]. VIs were calculated in R (R 3.5.1, R Foundation for Statistical Computing, Vienna, Austria) based on the mean value of the original reflectance of the spectral bands for each plot.

**Partial least square and random forest regression.**   Further analysis and model calculation were done with R. Two machine learning methods were used for model calibration by means of the *caret* package [57] for modelling FM and DM biomass and $N_{Fix}$: Partial Least Square (PLS) and Random Forest (RF) regression. PLS reduces the number of highly correlated independent variables (i.e. spectral bands) by linear combinations to a few latent vectors (principal components), which cover the maximum of covariance between independent and dependent variables (i.e. FM, DM, or $N_{Fix}$) to build a regression model [58]. Another machine learning algorithm, which in contrast observes non-linear relationships, is RF regression. RF, introduced by Breiman [59], builds multiple decision trees for regression with a random selection of sub-datasets as input variables. Both regression algorithms, PLS and RF, are common techniques for spectral analysis, including highly correlated independent variables, to estimate biomass yield and quality, also in the field of grassland and forage production [33,35,60].

For an internal training and validation of the PLS and RF models, a leave-one-out-cross-validation (LOOCV) was performed. For PLS the tuning parameter *ncomp*, which defines the number of principal components to be tested, was set to 10% of the number of samples with 4 as a minimum (for DM and FM: 19 for the whole dataset and 10 for both CG and LG; for $N_{Fix}$: 5 for the whole dataset and 4 for CG and LG). Two tuning parameters were set for RF in order

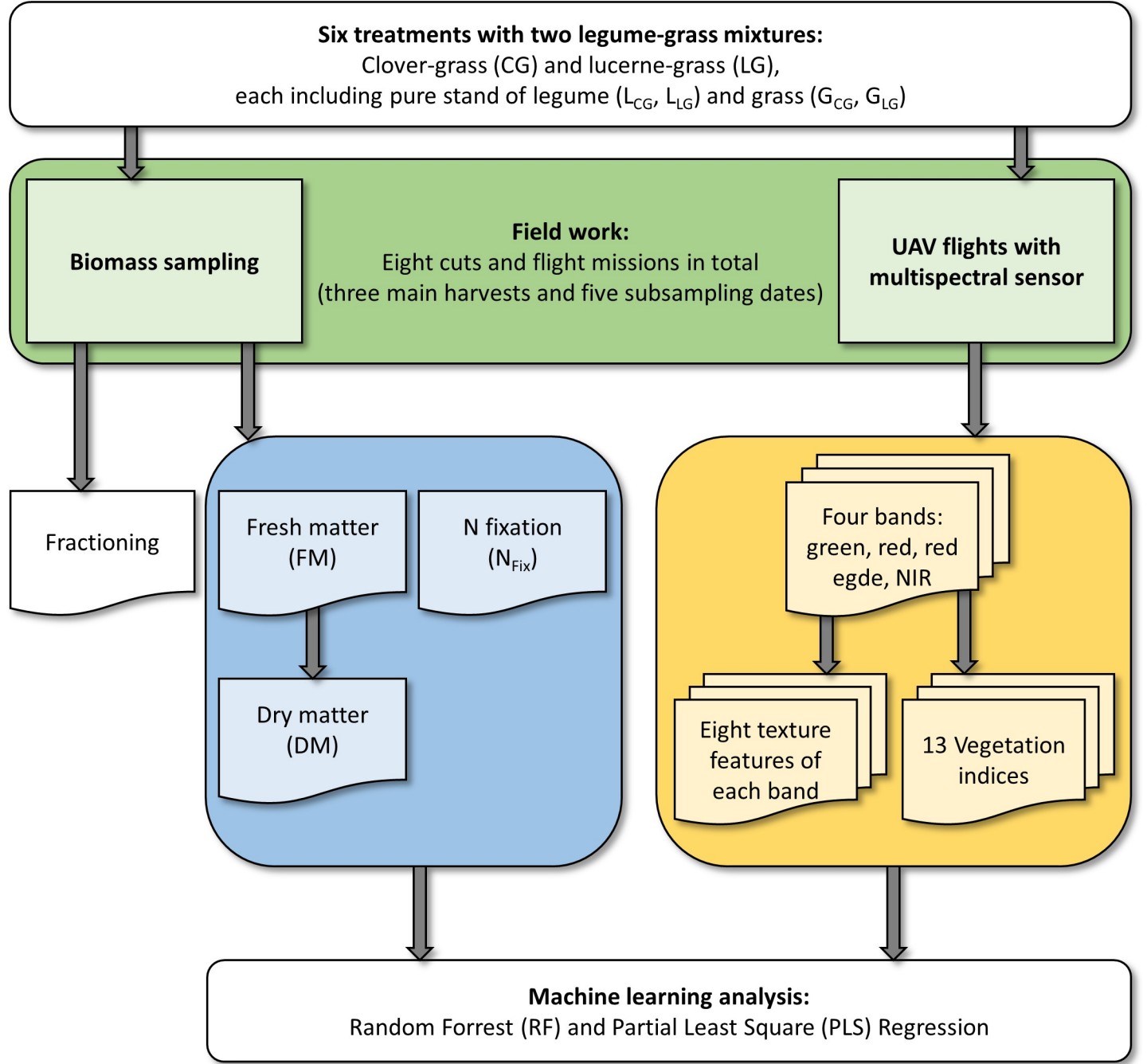

**Fig 2. Workflow.** Workflow of data acquisition and processing with fieldwork (green), dependent variables (blue) and independent variables (yellow) for data analysis.

to produce decision trees: *mtry*, which is the number of randomly selected variables and *ntree*, which represents the number of trees to grow. In this study, *mtry* was determined by the square root of the number of variables (here without texture: 5; with texture: 8) and *ntree* is automatically set to 500, as commonly recommended [61].

The four spectral bands together with eight texture parameters for each band and the 13 vegetation indices resulted in a total of 49 variables, which were used as independent variables,

**Table 1. Haralicks grey level co-occurrence matrix (GLCM) texture features.**

| Texture feature | Explanation (adapted from [54,56]) |
|---|---|
| 1. Energy | Measures the local steadiness of the grey levels |
| 2. Entropy | Measures randomness or degree of disorder |
| 3. Correlation | Shows the linear dependency of grey level values in the GLCM |
| 4. Inverse Difference Moment | Measures the local homogeneity |
| 5. Inertia | Measures the local contrast or amount of variations |
| 6. Cluster Shade | Measures skewness of the GLCM |
| 7. Cluster Prominence | Measures the asymmetry of the GLCM |
| 8. Haralick Correlation | Shows the probability of two pixels with similar grey level |

while FM, DM and $N_{Fix}$ were considered as dependent variables (Fig 2). To measure the accuracy of the PLS and RF models a cross-validation (CV) was implemented. Therefore, the whole dataset was divided into a training (75%) dataset for calibration of the model and a test dataset (25%). The test dataset included at least one datapoint of each treatment (CG, LG, $L_{CG}$, $L_{LG}$, $G_{CG}$, $G_{LG}$) and cut (three main harvests and five sub-sampling dates) for a later validation. To avoid bias by dividing the dataset, CV was run 100 times with randomly chosen training and test datasets. Performance of model prediction quality by CV was indicated by average coefficient of determination of the validation ($R^2_{val}$) (Eq 2), root mean square error of prediction (RMSEP) (Eq 3) and relative RMSEP (rRMSEP) (Eq 4).

$$R^2_{val} = \left[ 1 - \frac{\sum_{i=1}^n (y_i - \hat{y}_i)^2}{\sum_{i=1}^n (y_i - \bar{y}_i)^2} \right] \tag{2}$$

$$RMSEP = \sqrt{\frac{\sum_{i=1}^n (y_i - \hat{y}_i)^2}{n}} \tag{3}$$

$$rRMSEP = \frac{RMSEP}{\max(y_i) - \min(y_i)} \tag{4}$$

where $y_i$ is the measured variable (i. e. FM, DM, $N_{Fix}$), $\hat{y}_i$ is the predicted variable, $\bar{y}_i$ is the average measured variable and $n$ is the number of samples.

This was done for the whole dataset, as well as crop-specific for clover- and lucerne-grass, each including the mixture and the pure stands of the corresponding legumes and grasses. Furthermore, PLS and RF were compared with and without texture parameters and the best machine learning algorithm was selected based on the lowest median rRMSEP value for further analysis. To determine the variable importance for biomass and $N_{Fix}$ prediction, the best model out of the 100 cross-validated machine learning algorithms was identified based on the lowest rRMSE value.

## Results

Due to severe drought through the whole vegetation period, both biomass and N fixation were at a relatively low level. Nevertheless, the experimental set up with two legume-grass-mixtures (CG, LG) including the pure stands of legumes ($L_{CG}$, $L_{LG}$) and grass ($G_{CG}$, $G_{LG}$) as well as the various sampling dates through the vegetation period generated a wide range in biomass (Fig 3). In this multi-temporal study, the range of FM for CG and LG (0–100% legumes) was 0.38–33.95 and 0.23–28.56 t ha$^{-1}$, whereas the range of DM was 0.07–5.41 and 0.07–5.33 t ha$^{-1}$,

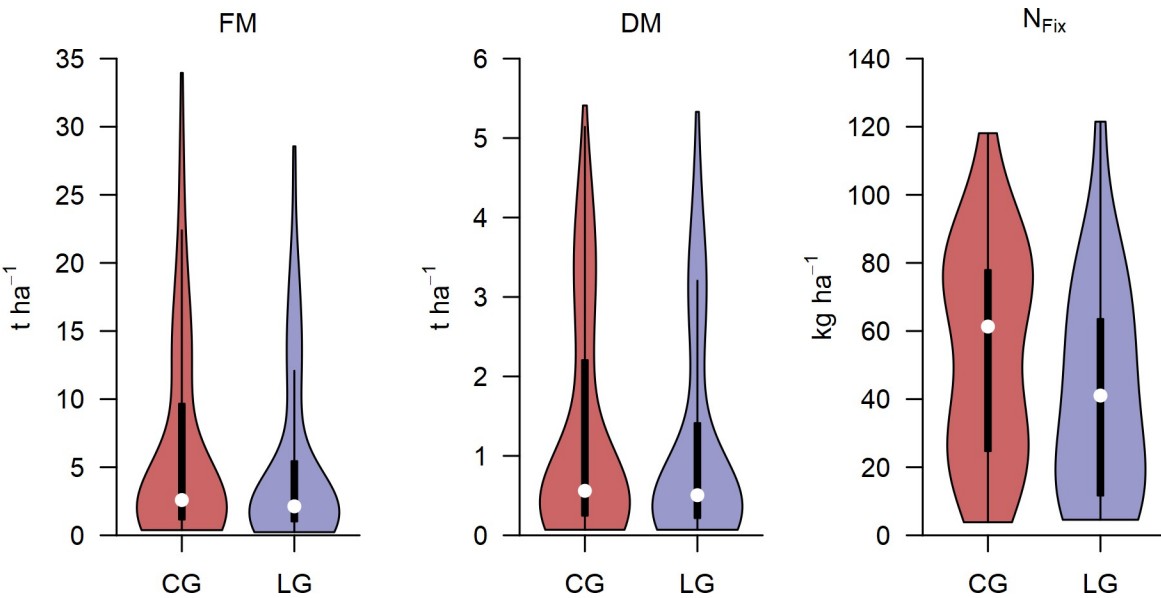

**Fig 3. Violin plot of ground truth data.** Violin plots of measured fresh (FM) and dry matter (DM) biomass and fixed N ($N_{Fix}$) of clover-(CG) and lucerne-grass (LG) mixtures including pure stands of legumes and grasses. Biomass data (FM, DM) were obtained at 3 main harvests and 5 sub-sampling dates, whereas $N_{Fix}$ was determined only at main harvests. Black boxes show the 25 and 75% percentile, white circles indicate the median, whiskers represent the 5 and 95% percentile, violins show point density.

respectively. $N_{Fix}$, which was calculated only for the main harvests, was 3.84–118.09 kg ha$^{-1}$ for CG and 4.57–121.48 kg ha$^{-1}$ for LG.

For every main harvest, the percentage of the different fractions (legumes, grass, herbs, senesced material) in the mixtures was determined to point out differences in the proportions of legumes (Fig 4). There were only slight differences between the two treatments CG and LG, but they differed between the three cuts. The proportion of legumes accounted for over 80% in

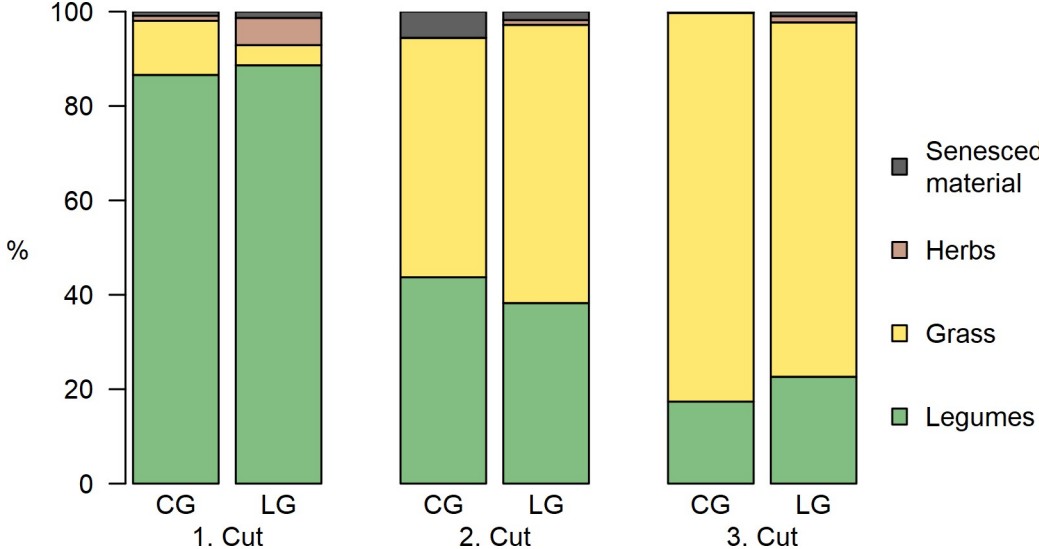

**Fig 4. Fractions of the mixtures.** Dry matter (DM) contribution of the legumes, grass, herbs and senesced material fractions for the three main cuts in the clover- (CG) and lucerne-grass (LG) mixtures.

the first cut and decreased for the second cut to about 40% and to 20% in the third cut. In contrast, the percentage of grass increased for every cut. Proportions of herbs and senesced material was negligible.

## Prediction models

Prediction models for biomass and $N_{Fix}$ were built with four spectral bands as well as 13 VIs and were compared with models including additional eight texture features for each band. Separate models were developed for the whole dataset (CG, LG, $G_{CG}$, $G_{LG}$, $L_{CG}$, $L_{LG}$) as well as specifically for the two legume-grass mixtures, including the corresponding pure stands of legumes and grass to cover the whole range of legume proportions occurring in farming practice (Table 2).

Inclusion of texture features generally improved cross validation results of both FM and DM models (Table 2). The lowest rRMSEP was found in the RF models with the whole dataset, where texture features reduced the relative error from 15.22 to 9.76% rRMSEP ($R^2_{val}$ = 0.62 to 0.86). For the crop-specific RF models texture features improved rRMSEP from 19.21 to 14.08% ($R^2_{val}$ = 0.55 to 0.76) for CG and from 19.04 to 14.62% ($R^2_{val}$ = 0.54 to 0.74) for LG. The same trends were obtained from the PLS models, but rRMSEP values were 2–3% higher (Table 2).

In general, $N_{Fix}$ prediction showed no consistent improvement and preferable model algorithm by the integration of texture features. For RF with LG, where rRMSEP was reduced from 26.33 to 23.75%, although it could not outperform the PLS model (Table 2). In contrast CG showed the best model by RF without texture (rRMSEP = 25.39%). The lowest rRMSEP of 17.88% was found for the whole dataset by RF with texture features, whereas rRMSEP excluding texture was 17.91%. Considering the crop-specific models, the best performance was found for PLS with texture, where the model for LG (rRMSEP = 20.90%; $R^2$ = 0.80) performed better than for CG without texture features (rRMSEP = 24.46%; $R^2$ = 0.69).

**Table 2. Cross-validation results.**

| Variable | Texture (T) | Model | Whole dataset | | | | Clover-grass (CG, $L_{CG}$, $G_{CG}$) | | | | Lucerne-grass (LG, $L_{LG}$, $G_{LG}$) | | | |
| | | | n | $R^2_{val}$ | RMSEP | rRMSEP | n | $R^2_{val}$ | RMSEP | rRMSEP | n | $R^2_{val}$ | RMSEP | rRMSEP |
|---|---|---|---|---|---|---|---|---|---|---|---|---|---|---|
| FM (t ha⁻¹) | Without T | PLS | 192 | 0.46 | 5.18 | 18.54 | 96 | 0.44 | 5.90 | 21.51 | 96 | 0.42 | 4.85 | 21.69 |
| | | RF | 192 | 0.62 | 4.23 | 15.22 | 96 | 0.55 | 5.32 | 19.21 | 96 | 0.54 | 4.52 | 19.04 |
| | With T | PLS | 192 | 0.77 | 3.60 | 12.13 | 96 | 0.70 | 4.30 | 15.75 | 96 | 0.70 | 3.56 | 16.58 |
| | | RF | 192 | **0.86** | **2.79** | **9.76** | 96 | **0.76** | **3.95** | **14.08** | 96 | **0.74** | **3.27** | **14.62** |
| DM (t ha⁻¹) | Without T | PLS | 192 | 0.53 | 0.96 | 20.05 | 96 | 0.51 | 1.03 | 21.88 | 96 | 0.49 | 0.96 | 21.94 |
| | | RF | 192 | 0.62 | 0.86 | 17.30 | 96 | 0.56 | 0.98 | 21.28 | 96 | 0.55 | 0.93 | 20.41 |
| | With T | PLS | 192 | 0.76 | 0.72 | 15.09 | 96 | 0.66 | 0.89 | 18.68 | 96 | 0.68 | 0.78 | 19.04 |
| | | RF | 192 | **0.87** | **0.52** | **10.78** | 96 | **0.79** | **0.71** | **15.05** | 96 | **0.77** | **0.66** | **15.28** |
| $N_{Fix}$ (kg ha⁻¹) | Without T | PLS | 48 | 0.72 | 18.12 | 18.98 | 24 | **0.69** | **19.87** | **24.46** | 24 | 0.81 | 18.56 | 22.25 |
| | | RF | 48 | 0.74 | 17.01 | 17.91 | 24 | 0.73 | 18.46 | 25.39 | 24 | 0.60 | 20.03 | 26.33 |
| | With T | PLS | 48 | 0.70 | 17.86 | 19.56 | 24 | 0.49 | 27.73 | 36.86 | 24 | **0.83** | **19.86** | **20.90** |
| | | RF | 48 | **0.76** | **16.77** | **17.88** | 24 | 0.67 | 18.99 | 26.41 | 24 | 0.79 | 17.34 | 23.75 |

Median value of 100 randomly executed cross-validations of Partial Least Square (PLS) and Random Forest (RF) regression including multispectral variables and indices with and without texture features (T) for fresh (FM) and dry matter (DM) as well as fixed N ($N_{Fix}$) for the whole dataset as well as crop-specific: clover-grass (CG) and lucerne-grass (LG) mixtures including the pure stands of legumes ($L_{CG}$, $L_{LG}$) and grass ($G_{CG}$, $G_{LG}$); n = number of datapoints, $R^2_{val}$ = coefficient of determination of validation, RMSEP = root mean square error of prediction, rRMSEP = relative RMSEP (%).

The best prediction model algorithm (i. e. RF or PLS) was chosen by lowest rRMSEP value for further assessment. Coefficient of determination of validation ($R^2_{val}$) for the best models varied between 0.69 and 0.87. The plots of fit in Fig 4 for measured versus predicted biomass and $N_{Fix}$ show the 100 times randomly repeated prediction models for the whole dataset as well as crop-specifically. As the validation dataset contained at least one data point from each treatment and cut, a wide range of crop conditions was covered by the models. Despite of that all models showed an underestimation at higher biomass levels (Fig 5), while this trend was not as clearly visible for $N_{Fix}$.

To identify the contribution of texture features compared to single bands and VIs in the developed prediction models, the variable importance was calculated (Fig 6). Therefore, out of the 100 CV-models the best model based on the lowest rRMSEP was chosen and compared with and without texture for the whole dataset (Fig 6). For FM without texture features the three most important variables were MCARI, NDRE and GCI, which changed with the inclusion of texture features to G_Tex_3, R_Tex_7 and CVI. A similar trend is visible for DM, where GCI, GNDVI and R were the most important variables, whereas GCI, R_Tex_7 and G_Tex_3 contributed the most to the model when texture features were included. Particularly texture features of the red band ranked relatively high, which also applied for crop-specific models (S2 and S3 Figs). For $N_{Fix}$ without texture features the three most important variables were RE, NIR and CVI, which changed to RE, NIR_Tex_8 and RE_Tex_8 after including texture features.

For practical farming the accuracy of total annual biomass is relevant, which is the cumulated biomass of the main harvests of one year. Therefore, biomass (FM, DM) and $N_{Fix}$ measured and predicted by the best prediction algorithm for the two mixtures CG and LG is shown in Fig 7 with cumulated data. Though the two legume-grass mixtures were on different biomass and $N_{Fix}$ levels, they showed very similar patterns concerning the trend of observed and predicted values. In total after the third harvest (H3) predicted FM and DM was underestimated by 1.18 and 1.25 t ha$^{-1}$ for CG as well as 1.52 and 2.38 t ha$^{-1}$ for LG respectively. $N_{Fix}$ was overestimated at all cuts with annual $N_{Fix}$ (H3) was overestimated by 13.69 kg ha$^{-1}$ for CG and by 9.96 kg ha$^{-1}$ for LG.

## Discussion

The aim of this study was to provide aboveground biomass and $N_{Fix}$ estimation models in two legume-grass mixtures through a whole vegetation period based on multispectral information. The prediction models covered different proportions of legumes to represent the variable conditions in practical farming and the multi-temporal data acquisition offered a wide range of biomass levels, which is a prerequisite for robust and generalised models [62]. All multispectral information, containing also VIs and texture features, arose from four spectral bands. Consequently, no additional sensors were needed, which reduced measurement errors [63] and makes this method time and cost efficient.

The first specific objective of this study was the development of biomass and $N_{Fix}$ prediction models for two common legume-grass mixtures. Though differences in model performance between the two machine learning algorithms (i.e. PLS and RF) were minor, biomass prediction by RF performed better (in terms of rRMSE) than by PLS both for the whole as well as for the crop-specific dataset. Similar findings were reported by Zhou et al. [64], who found Support Vector Machine (SVM) to perform better than PLS with hyperspectral reflectance data captured with a radiometer (400–1000 nm) in different legume-grass mixtures. The authors pointed out, that there might exist a non-linear relationship between spectral information and biomass, which could not be captured by PLS modelling. For $N_{Fix}$ the best model for the whole

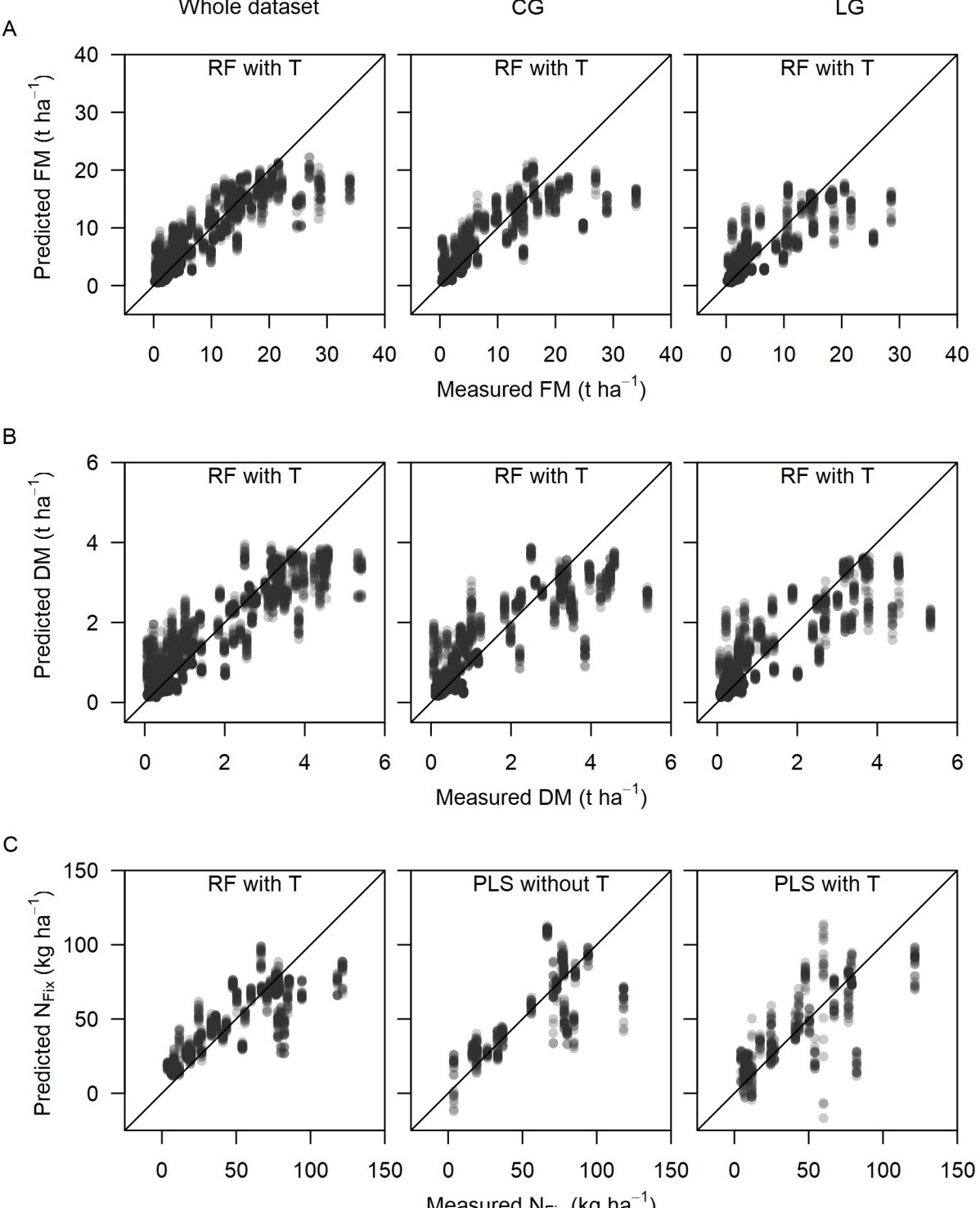

**Fig 5. Plot of fit.** Scatterplot for measured and predicted fresh (FM) (A) and dry matter (DM) (B) and fixed N ($N_{Fix}$) (C) for the whole dataset as well as crop-specific: clover-grass (CG) and lucerne-grass (LG) mixtures including the pure stands of legumes and grass. Plots show the best prediction algorithm, Partial Least Square or Random Forest (RF), with 100 randomly selected test and training data sets based on data from 3 main harvests and 6 sub-sampling dates, whereas $N_{Fix}$ contains only main harvests.

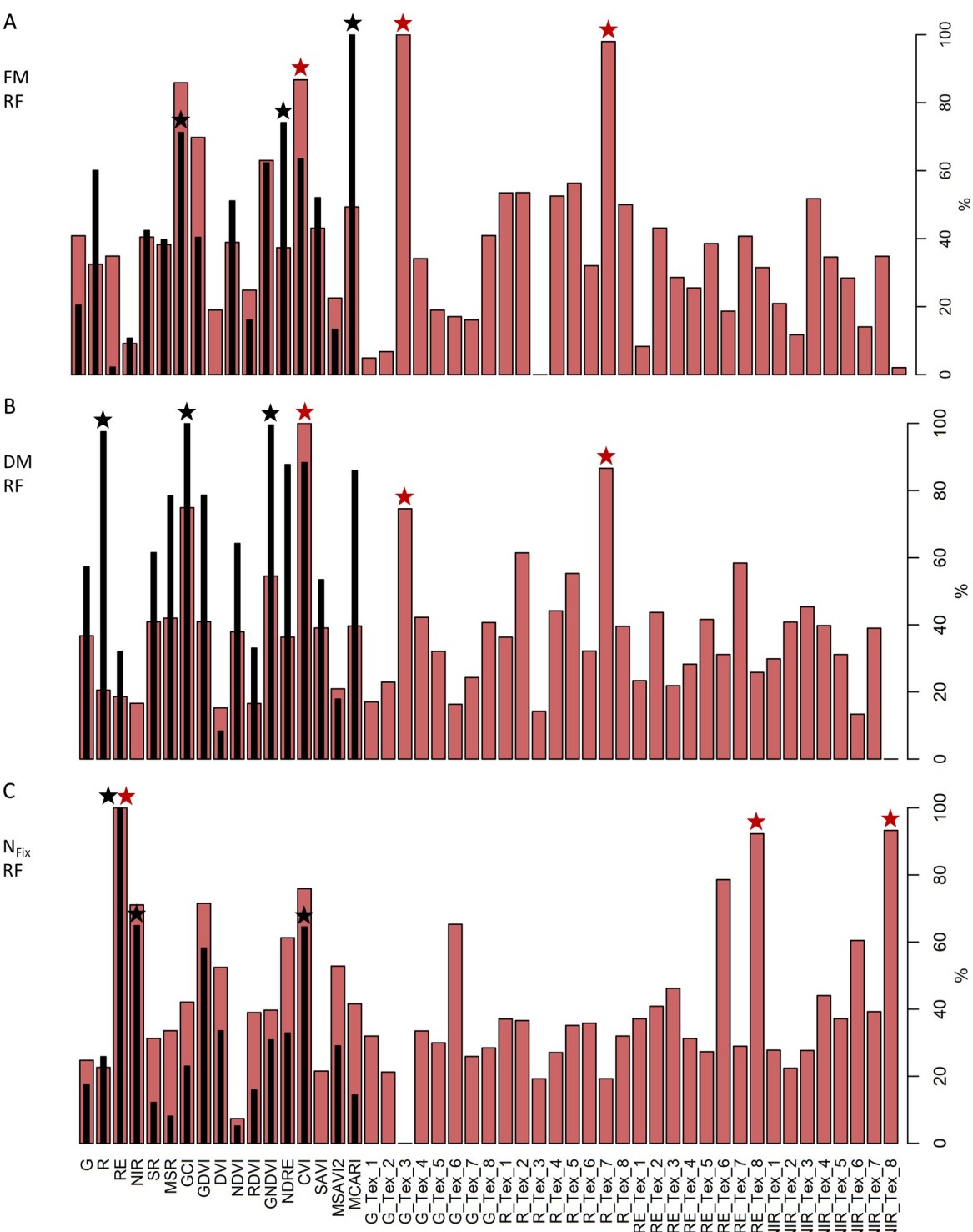

**Fig 6. Variable importance.** Variable importance of prediction models for fresh (FM) (A) and dry matter (DM) (B) as well as fixed N (N$_{Fix}$) (C) for the whole dataset built with four spectral bands, 13 vegetation indices and with (red) and without (black) 8 texture features of each band. Stars indicate the three highest rankings of variables with (red) and without texture features (black) in the model. Plots show the best prediction algorithm, Partial Least Square (PLS) or Random Forest (RF), with the best of 100 randomly selected test and training data sets based on data from 3 main harvests and 6 sub-sampling dates, whereas N$_{Fix}$ contains only main harvests.

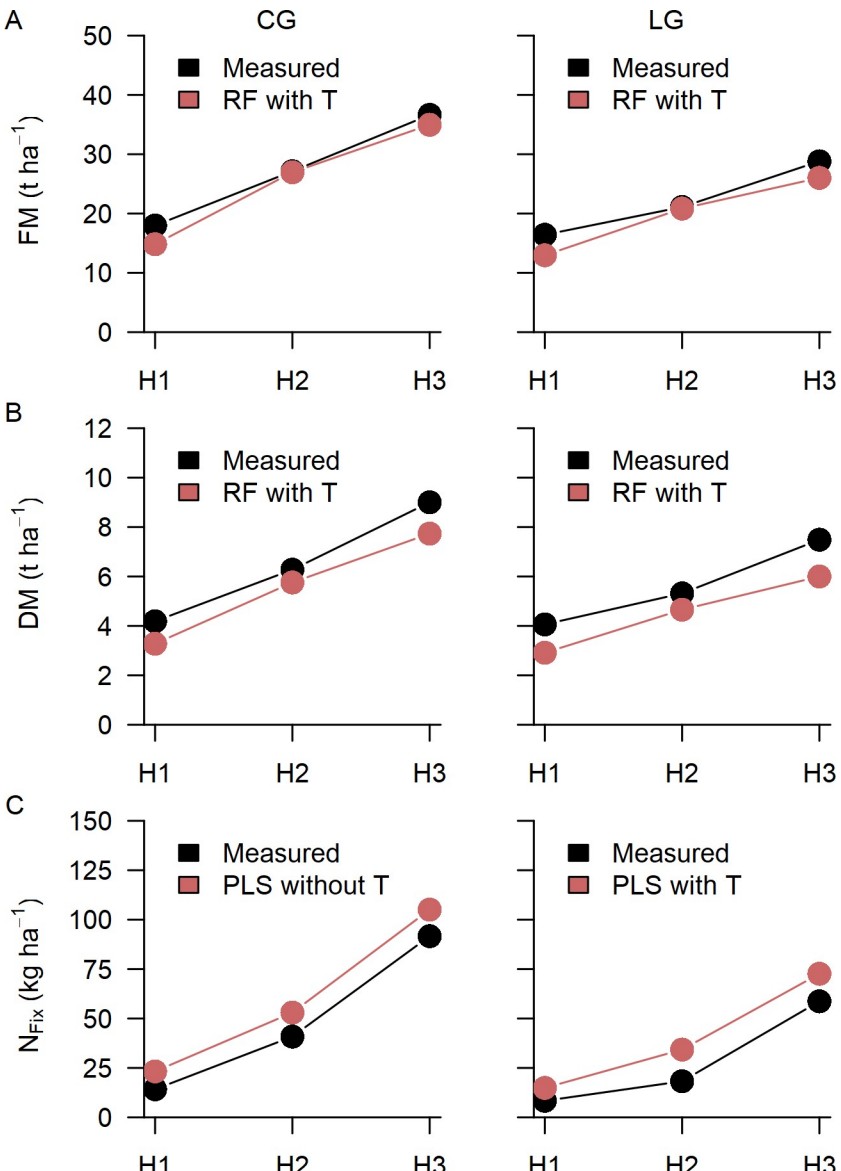

**Fig 7. Annual biomass.** Measured (black) and predicted (red) cumulated biomass for fresh (FM) (A) and dry matter (DM) (B) as well as fixed N ($N_{Fix}$) (C) for clover-grass (CG) and lucerne-grass (LG) mixtures, after the first (H1), second (H2: H1+H2) and third (H3: H1+H2+H3) cut. Best machine learning algorithm is shown of Partial Least Square (PLS) and Random Forest (RF) regression with and without texture features (T).

dataset was given by RF, whereas the crop-specific regression analysis was better with PLS. Morin et al. [55] stated that RF showed weaknesses in dealing with small sample sizes. In our study the dataset for model calibration of $N_{Fix}$ was limited to three main harvests only, which may have decreased model robustness.

In contrast to findings by Grüner et al. [24], where biomass prediction models by UAV height measurement showed substantial crop-specific differences, no clear differences were found in this study between clover- and lucerne-grass mixtures. Interestingly, for biomass and $N_{Fix}$ the best models with an average rRMSEP of 10% (FM), 11% (DM) and 18% ($N_{Fix}$) were obtained based on the whole dataset, which produced rather universal and crop-unspecific

models. Model validation by a preferably independent test dataset is desirable for the assessment of prediction accuracy. However, our models were created based on datasets from one experimental site only and a limited variety of plant species which may limit the transfer to other locations and time periods [45].

$N_{Fix}$ models were less accurate with a relative error of 18–24%. To our best knowledge, no other studies investigated the relationships between spectral information and N fixation by legume-grass mixtures. The relationships between $N_{Fix}$ and multispectral information with $R^2$ from 0.69 to 0.84 found in our study must be interpreted with caution, as different impacts of N flux in the soil, air and plant affect N fixation of forage legumes in mixtures with grasses [16]. Knoblauch et al. [65] could not capture N dynamics of the soil by a multispectral sensor with two bands (red, NIR) or by calculating NDVI and SR (Simple Ratio) in a greenhouse experiment with perennial ryegrass (*Lolium perenne* L.). The authors concluded that the above mentioned effects cause problems to measure $N_{Fix}$ so far by optical remote sensing, as it only obtains spectral information of the biomass and the top soil surface, but furthermore, does not consider the root tissue which also contains atmospherically fixed N [7]. $N_{Fix}$ model accuracy may also depend on the methodology for measuring reference data. $N_{Fix}$ calculation by the difference method, which is widely-spread and relatively simple to use, assumes that grass and legumes have the same intensity of soil N absorption. However, grass usually absorbs more soil N, which leads to an underestimation of the method [7]. For this reason, Thilakaratha et al. [16] suggested to include non-N-fixing legumes of the same species as a reference, not only for a more precise difference method, but also to calibrate the spectral information and to use it for an improved classification of N fixation.

The second and third objective of our study was to compare the performance of the biomass and $N_{Fix}$ prediction models with and without texture features and to identify the importance of the variables. Models for FM and DM showed poor accuracies without texture features with an rRMSEP between 15 to 22%. Due to missing rainfall through the whole vegetation period and as no fertilizer was applied, especially pure stands of grass were growing poorly after the first cut. Biewer et al. [66] found that models build with VIs solely (i. e. NDVI and SR) showed weaknesses in predicting mature and dry grass swards due to alteration of the spectral information. Though mixed stands with plants of varying chemical (e. g. pigments and water) and morphological traits (e. g. angle and structure of leaves) may show similar spectral reflection patterns [67], texture features may help to distinguish single plant species by considering the spatial heterogeneity [68]. Our results clearly demonstrated that inclusion of texture features reduced rRMSEP by almost the half (rRMSEP = 10–15%) and resulted in improved model accuracies ($R^2$ = 0.75–0.87). Furthermore, texture features showed high rankings in the variable importance of the models, especially for those of the red band. The latter finding is confirmed by the work of Gallardo-Cruz et al. [37] who found texture features from red and NIR bands as the most important in satellite-based remote sensing for vegetation classification and modelling of height and biodiversity.

Thus far, only one study included texture analysis for biomass prediction of agricultural crops. Using multispectral UAV images Zheng at al. [48] found rather poor relationships between different texture features and DM over two vegetations periods. However, model accuracy could be improved by using a new Normalized Difference texture index (NDTI) instead of single texture features ($R^2$ = 0.44–0.75 for the whole dataset). A comparison to legume-grass mixtures, which form more heterogeneous canopies, is difficult. Nevertheless, Zheng at al. point out that a normalization of texture information by an index excludes background noises like soil reflection and varying sun position and, thus, may improve biomass prediction in multi-temporal studies with legume-grass mixtures, but more research is required in this area.

$N_{Fix}$ was improved by texture features except for CG, although texture showed high rankings for variable importance. As $N_{Fix}$ depends on several different conditions (e.g. soil variability), this plant trait cannot be captured easily by spectral information of the plant canopy. Such limited information may be the reason for the inconsistent effects of texture features on model accuracy. A broader spectral range or hyperspectral information from additional sampling dates as well as from bare soil areas at the time of measurement, which act as a reference for the soil N content [69] may improve model accuracy.

The last objective of our study was the assessment of total annual biomass and $N_{Fix}$ for CG and LG. Total annuals represent the level at which farmers, plant breeders and advisory services usually conduct the evaluation of crop performance and where data are available as a basis for feeding plans for ruminants or biogas plants. Annual FM and DM was underestimated for both CG and LG by around 1 to 2 t ha$^{-1}$, with slightly better accuracy for FM. Surprisingly, a prediction of annual biomass by crop height, which was estimated from drone-based RGB imaging and photogrammetric analysis, showed greater potential [24]. Contrary, annual total $N_{Fix}$ for CG and LG was slightly overestimated by 10–14 kg ha$^{-1}$. Compared to data reported for CG and LG, which indicate a total annual N fixation of up to 300 kg ha$^{-1}$ [7,9,70], total annual $N_{Fix}$ in our study was generally on a relatively low level, which in combination with a small sample size may have created difficulties in the modelling process. Annual biomass and $N_{Fix}$ prediction by multispectral information, thus, should be considered as a first approach for the support of farm management decisions, which still needs further improvement.

In general, accurate biomass prediction by multispectral UAV data depends on several parameters during data acquisition, like recording time, image quality and model tuning. Though the sensor used in our study had an integrated sunshine sensor for automatic calibration of every picture, multi-temporal data acquisition requires stable and calm weather conditions and equal time points of flight missions [71], which was difficult to comply with even under experimental conditions and all the more so poses challenges under farming conditions. Changing atmospheric condition between the sampling dates might also affect data quality, which can only be compensated by proper atmospheric correction [72]. Nevertheless, atmospheric algorithms are complex and intense atmospheric measurements have to be done simultaneously for each flight, which increases measurement time [21,72,73]. Moreover, moving plant leaves by wind or blurred images need to be avoided to keep texture accuracy high [52]. As flight speed in our study was very slow (1 m/s) image quality was high and no image had to be excluded from the analysis. Especially for texture features, image quality and resolution play a decisive role [36], which depends on the sensor resolution in combination with the flight altitude. With the sensor used in our study a flight height of 50 and 20 m resulted in an image resolution of 2–4 cm, which was then resampled to 4.5 cm. Different ground resolutions should be avoided in future studies to keep unified conditions for data analysis As plant and especially grass leaves can be thinner than 2 cm, a higher spatial resolution may improve texture resolution and, therefore, biomass prediction accuracy. Furthermore, in our study only default parameter settings for texture processing were used. The impact and versatility of tuning parameters and window size for texture processing was shown elsewhere [36,47,55] and should be considered in future research to improve quality of biomass estimation in legume-grass mixtures. Further work is needed to assess the potential of this promising tool at other sites and different legume-grass mixtures.

## Conclusion

Non-destructive and fast aboveground biomass and $N_{Fix}$ prediction tools are desirable for practical farm management, especially for organic farmers who are depending on legumes as

an N source. For this purpose, the aim of this multi-temporal field study was to provide above-ground biomass and $N_{Fix}$ estimation models based on UAV-borne multispectral information. Two machine learning methods were tested (PLS and RF) using data from two different legume-grass mixtures and associated pure legume and grass through a whole vegetation period. We successfully developed a procedure for biomass prediction by inclusion of texture features from a grey level co-occurrence matrix. RF produced the best results for the whole dataset based on the two legume-grass mixtures including the pure stands of legumes and grass, both for biomass and $N_{Fix}$. Although prediction of fixed N seemed to be more complex, strong relationships were found between $N_{Fix}$ and multispectral information under field conditions. In conclusion, multispectral information including texture features from one single sensor on a UAV proved to be a very promising tool for biomass and $N_{Fix}$ prediction in legume-grass mixtures.

## Supporting information

**S1 Table. Treatments.** CG = Clover-grass; LG = Lucerne-grass.
(TIF)

**S2 Table. Vegetation indices.** Vegetation indices calculated with four bands captured by the multispectral sensor used in this study: green, red, red edge (RE) and near infra-red (NIR).
(TIF)

**S3 Table. Photogrammetric processing information.**
(TIF)

**S1 Fig. Model accuracy.** Boxplots for the model accuracy created by 100 cross-validations for the whole dataset as well as crop-specific: clover-grass (CG) and lucerne-grass (LG) mixtures including the pure stands of legumes and grass. Plots show the best prediction algorithm, with 100 randomly selected test and training data sets based on data from 3 main harvests and 6 sub-sampling dates, whereas $N_{Fix}$ contains only main harvests. Boxes show the 25 and 75% percentile, the solid line indicates the median, the whiskers represent the 5 and 95% percentile, circles show outliers.
(TIFF)

**S2 Fig. Variable importance for clover-grass.** Variable importance of prediction models for fresh (FM) (A) and dry matter (DM) (B) as well as fixed N ($N_{Fix}$) (C) for the whole dataset built with four spectral bands, 13 vegetation indices and with (red) and without (black) 8 texture features of each band. Stars indicate the three highest rankings of variables with (red) and without texture features (black) in the model. Plots show the best prediction algorithm, Partial Least Square (PLS) or Random Forest (RF), with the best of 100 randomly selected test and training data sets based on data from 3 main harvests and 6 sub-sampling dates, whereas $N_{Fix}$ contains only main harvests.
(TIF)

**S3 Fig. Variable importance for lucerne-grass.** Variable importance of prediction models for fresh (FM) (A) and dry matter (DM) (B) as well as fixed N ($N_{Fix}$) (C) for the whole dataset built with four spectral bands, 13 vegetation indices and with (red) and without (black) 8 texture features of each band. Stars indicate the three highest rankings of variables with (red) and without texture features (black) in the model. Plots show the best prediction algorithm, Partial Least Square (PLS) or Random Forest (RF), with the best of 100 randomly selected test and training data sets based on data from 3 main harvests and 6 sub-sampling dates, whereas $N_{Fix}$

contains only main harvests.
(TIF)

## Acknowledgments

The authors would like to thank Wolfgang Funke for his support in field data collection and Rüdiger Graß for advice in crop management.

## Author Contributions

**Conceptualization:** Michael Wachendorf, Thomas Astor.

**Formal analysis:** Esther Grüner.

**Investigation:** Esther Grüner.

**Methodology:** Esther Grüner.

**Supervision:** Michael Wachendorf, Thomas Astor.

**Visualization:** Esther Grüner.

**Writing – original draft:** Esther Grüner.

**Writing – review & editing:** Michael Wachendorf, Thomas Astor.

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
