## [Decision Letter · Decision Letter 0]

5 Mar 2020

PONE-D-19-34750

The potential of UAV-borne spectral and textural information for predicting aboveground biomass and N fixation in legume-grass mixtures

PLOS ONE

Dear Mrs. Grüner,

Thank you for submitting your manuscript to PLOS ONE. After careful consideration, we feel that it has merit but does not fully meet PLOS ONE’s publication criteria as it currently stands. Therefore, we invite you to submit a revised version of the manuscript that addresses the points raised during the review process.

Please make sure you address all issues rised by the reviewers, especially focusing on Rew. 1 comments, clarifying the methods and providing both metadata and orthomosaics in supplements.

We would appreciate receiving your revised manuscript by Apr 19 2020 11:59PM. To enhance the reproducibility of your results, we recommend that if applicable you deposit your laboratory protocols in protocols.io, where a protocol can be assigned its own identifier (DOI) such that it can be cited independently in the future. For instructions see: http://journals.plos.org/plosone/s/submission-guidelines#loc-laboratory-protocols

We look forward to receiving your revised manuscript.

Kind regards,

Jana Müllerová, Ph.D

Academic Editor

PLOS ONE

Journal Requirements:

2. Our internal editors have looked over your manuscript and determined that it is within the scope of our Plant Phenomics & Precision Agriculture Call for Papers. This collection of papers is headed by a team of Guest Editors for PLOS ONE. The Collection will encompass a diverse range of research articles spanning disciplines, methods and applications.  Additional information can be found on our announcement page: https://plos.io/phenomics.

If you would like your manuscript to be considered for this collection, please let us know in your cover letter and we will ensure that your paper is treated as if you were responding to this call. If you would prefer to remove your manuscript from collection consideration, please specify this in the cover letter.

4. Please include your tables as part of your main manuscript and remove the individual files. ** Please note that supplementary tables (should remain/ be uploaded) as separate "supporting information" files **.

Reviewers' comments:

Reviewer's Responses to Questions

**Comments to the Author**

1. Is the manuscript technically sound, and do the data support the conclusions?

Reviewer #1: Partly

Reviewer #2: Yes

2. Has the statistical analysis been performed appropriately and rigorously? 

Reviewer #1: Yes

Reviewer #2: Yes

3. Have the authors made all data underlying the findings in their manuscript fully available?

Reviewer #1: No

Reviewer #2: Yes

4. Is the manuscript presented in an intelligible fashion and written in standard English?

Reviewer #1: Yes

Reviewer #2: Yes

5. Review Comments to the Author

Reviewer #1: 1. Is the manuscript technically sound, and do the data support the conclusions?

Overall, the manuscript is technical sound and an interesting contribution to the field. The authors have performed a thorough data capture of field data and reference data as well. I would to suggest to review the following sections in order to provide a clarification:

Materials and Methods

A lot of data have been captured (multispectral UAS, Field Reference, GCP) and a lot of different variables have been computed and statistically analyzed. It would be very helpful to provide a summary table about all available data sets, calculated variables (VI, GLCM) and performed regression analysis.

165: Please make sure that the full experimental design necessary and relevant in order to follow your argumentation is summarized in this manuscript. It seems to me that this is described between 165 -174, please review this issue.

189 +: Fig. 1: From this figure it is not clear, how B (i.e. the design of the experimental Layout) can be seen in A? I would suggest to clearly indicate the layouts from B in A.

UAV Image acquisition and data preprocessing.

This chapter has significant potential for improvement. It is not clear why the UAS was flown manually and not using a predefined flight mission plan. This brings a certain level of uncertainty regarding data quality of UAS imagery as you always have different flight geometries. Furthermore, it is not clear why there has been a change of flight altitude between the first 2 flights and the 3rd flight resulting in different GSD’s for the different flights. Please argue, why flight 3 was included in the analysis as I assume a change in resolution will also affect for example the calculation of GLCM?

In order make the photogrammetric processing transparent, I suggest to add as supplementary information the metadata from the processing reports, i.e. in particular the overlap, and also the RMSE as the authors have used GCP’s.

In general, please add a comment what is the desired GSD in order to capture the desired grassland properly. How is this reflected in selecting the flight altitude in combination with sensor resolution.

Please describe also, if, and if yes, how you performed a resampling as each UAS orthomosaic has a different resultion – what was here your final target resolution in order perform a proper coregistration of the individual images from each mission.

It would also strengthen your manuscript, if you can provide the orthomosaics for all three missions.

Data analysis and machine learning

245: Please explain, what do you mean by “avoiding the first 1.5 m”. In 167 you describe the size of a field plot is 1.5mx12m – if avoid 1.5 m – what is then left from the plot? Please clarify! Maybe a figure showing your analysis design in detail might help?

A zonal statistic has been performed calculating the mean of each variable. I would suggest to provide here a basic statistic description of the variables including MIN, Max, STDV etc. At least I would suggest to add here a comment on variability of the individual variables and the potential of loosing information due to averaging?

Texture features of images

251-252: Please explain, what is the rationale behind selecting 8 textural features out of the 14 as suggested by Haralick? Is there any evidence which of selected 8 is suitable for grassland texture? Please justify/argue/clarify your selection of texture features.

Vegetation index calculation

Please argue, why you have selected the mentioned 13 VI?

2. Has the statistical analysis been performed appropriately and rigorously?

Yes.

3. Have the authors made all data underlying the findings in their manuscript fully available?

Not for the review.

4. Is the manuscript presented in an intelligible fashion and written in standard English?

Yes.

158: ……located in northern Hessen.

Reviewer #2: The manuscript is very well organised and the English style is good. The topic is interesting for the potential impact on operational precision farming. I have just few concerns:

1) It seems that the multispectral images were calibrated in reflectance using the irradiance sensor installed on the UAV. If this is true, then your images were calibrated in reflectance at platform level and not in ground reflectance. Since you are processing a time series of images this could be an important aspect to discuss. Also VI can be affected by the atmosphere, depending on their formulation. I think that this argument deserves a discussion;

2) In you analyses you are using the 13 VI as independent variables. However I would like to stress that these VI are usually correlated and for sure they are correlated with the original bands values. Please confirm or not that this is not a problem for the validity of your application, from statistical point of view.

Pag 17 - please check this sentence "The radiometric resolution was set to 0 to 255."

6. PLOS authors have the option to publish the peer review history of their article (what does this mean?). If published, this will include your full peer review and any attached files.

Reviewer #1: No

Reviewer #2: No

---

## [Author Response · Author response to Decision Letter 0]

31 Mar 2020

Manuscript ID: PONE-D-19-34750 - Response letter

Dear Mrs. Müllerová,

Dear Reviewers,

thank you very much for the review of our manuscript titled „ The potential of UAV-borne spectral and textural information for predicting aboveground biomass and N fixation in legume-grass mixtures “, which greatly helped to improve it. We have carefully implemented the comments. 

Due to modification of the data processing (as proposed by Reviewer 1), major changes were made in section “UAV image acquisition and data pre-processing” (from L: 222), “Prediction models” (from L: 403) and minor changes in section “Discussion” (from L: 578).

Providing metadata (140 data point with 52 variables) and eight orthomosaics to the supplements of this manuscript seems not operable to us. This data will be part of the published data in a public repository.

Please find our specific answer to each reviewer’s comments below in bold letters.

Yours sincerely

Esther Grüner

 

PLOS ONE

Journal Requirements:

2. Our internal editors have looked over your manuscript and determined that it is within the scope of our Plant Phenomics & Precision Agriculture Call for Papers. This collection of papers is headed by a team of Guest Editors for PLOS ONE. The Collection will encompass a diverse range of research articles spanning disciplines, methods and applications. Additional information can be found on our announcement page: https://plos.io/phenomics.

If you would like your manuscript to be considered for this collection, please let us know in your cover letter and we will ensure that your paper is treated as if you were responding to this call. If you would prefer to remove your manuscript from collection consideration, please specify this in the cover letter.

Yes, we would like our manuscript to be considered for Plant Phenomics & Precision Agriculture Call for Papers. Thank you.

Metadata and orthomosaics will be published at gfbio (German Federation for Biological Data: www.gfbio.org), which is in progress.

4. Please include your tables as part of your main manuscript and remove the individual files. ** Please note that supplementary tables (should remain/ be uploaded) as separate "supporting information" files **.

Done.

 

Reviewers' comments:

Reviewer's Responses to Questions

Comments to the Author

1. Is the manuscript technically sound, and do the data support the conclusions?

Reviewer #1: Partly

Reviewer #2: Yes

2. Has the statistical analysis been performed appropriately and rigorously? 

Reviewer #1: Yes

Reviewer #2: Yes

3. Have the authors made all data underlying the findings in their manuscript fully available?

Reviewer #1: No

Reviewer #2: Yes

4. Is the manuscript presented in an intelligible fashion and written in standard English?

Reviewer #1: Yes

Reviewer #2: Yes

Reviewer #1: 1. Is the manuscript technically sound, and do the data support the conclusions?

Overall, the manuscript is technical sound and an interesting contribution to the field. The authors have performed a thorough data capture of field data and reference data as well. I would to suggest to review the following sections in order to provide a clarification:

Materials and Methods

A lot of data have been captured (multispectral UAS, Field Reference, GCP) and a lot of different variables have been computed and statistically analyzed. It would be very helpful to provide a summary table about all available data sets, calculated variables (VI, GLCM) and performed regression analysis.

The authors agree, that a lot of different variables were included in the analysis. Therefore, a workflow (Fig. 2) for a better overview was designed and implemented to the section “Experimental site and ground truth data” (L. 299).

165: Please make sure that the full experimental design necessary and relevant in order to follow your argumentation is summarized in this manuscript. It seems to me that this is described between 165 -174, please review this issue.

The authors ensured that all relevant information of the experimental design is included in section “Experimental site and ground truth data”. Minor changes were done (L. 173).

189 +: Fig. 1: From this figure it is not clear, how B (i.e. the design of the experimental Layout) can be seen in A? I would suggest to clearly indicate the layouts from B in A.

The authors decided to keep two separate figures for a better overview. The figure (Fig. 1, L.: 188) has been modified for a better understanding how B is included in A by white borders of the plots in A.

UAV Image acquisition and data preprocessing.

This chapter has significant potential for improvement. It is not clear why the UAS was flown manually and not using a predefined flight mission plan. This brings a certain level of uncertainty regarding data quality of UAS imagery as you always have different flight geometries. Furthermore, it is not clear why there has been a change of flight altitude between the first 2 flights and the 3rd flight resulting in different GSD’s for the different flights. Please argue, why flight 3 was included in the analysis as I assume a change in resolution will also affect for example the calculation of GLCM?

There are eight flight missions in total and two of them are done with a flight altitude of 50 m, whereas the remaining five are at 20 m. For these two flight missions the image overlap was 100%. Therefore, as a compromise between flight height and time, remaining cuts were flown at 20 m. All flight missions were flown manually as by removing the original camera, automatic flight missions were not possible due to internal technical problems. This information was included from L. 249 to 255. All flights were included in the analysis to keep data size high. This has only minor effects on spectral information, but the authors agree that it can affect texture analysis. Future work should focus on unified flight heights, which is now mentioned in section “Discussion” (L. 697). 

In order make the photogrammetric processing transparent, I suggest to add as supplementary information the metadata from the processing reports, i.e. in particular the overlap, and also the RMSE as the authors have used GCP’s.

Done (S3 Table, L.: 965). The information about the number of overlapping images is given in L. 263.

In general, please add a comment what is the desired GSD in order to capture the desired grassland properly. How is this reflected in selecting the flight altitude in combination with sensor resolution.

GSD had minor effects on multispectral information. Our aim was to keep the flight missions feasible and, therefore, it was a compromise between flight time and resolution. This was implemented in L. 250.

Please describe also, if, and if yes, how you performed a resampling as each UAS orthomosaic has a different resultion – what was here your final target resolution in order perform a proper coregistration of the individual images from each mission.

The authors thank the editor for this comment and agree, that a resampling is necessary. The orthomosaics were resampled to a unified resolution of 4.5 cm (L. 280). Consequently, all analysis was repeated and results changed minor. 

It would also strengthen your manuscript, if you can provide the orthomosaics for all three missions.

As there are eight orthomosaics in total (three harvests and five sub-sampling dates), such attachment would not substantially improve information in this manuscript. We suggest instead, that the orthomosaics will be part of the published data in a public repository.

Data analysis and machine learning

245: Please explain, what do you mean by “avoiding the first 1.5 m”. In 167 you describe the size of a field plot is 1.5mx12m – if avoid 1.5 m – what is then left from the plot? Please clarify! Maybe a figure showing your analysis design in detail might help?

To avoid disturbance of the sub-sampling in regrowth of biomass for the main harvests, the sub-sampling was restricted to the first 1.5 m (Fig. 1 B, light red area). Therefore, the remaining area for the analysis was 15.75 m2. This information was added in L. 290.

A zonal statistic has been performed calculating the mean of each variable. I would suggest to provide here a basic statistic description of the variables including MIN, Max, STDV etc. At least I would suggest to add here a comment on variability of the individual variables and the potential of loosing information due to averaging?

The authors agree, that by averaging variable values information may get lost. Nevertheless, by including texture features, spatial information is still part of the analysis. Due to the size and complexity the authors would like to avoid adding the statistical description (MIN, MAX, STDV etc.) for the 52 variables included in the study.

Texture features of images

251-252: Please explain, what is the rationale behind selecting 8 textural features out of the 14 as suggested by Haralick? Is there any evidence which of selected 8 is suitable for grassland texture? Please justify/argue/clarify your selection of texture features.

As the utilization of texture features was never done before in biomass estimation for grassland or legume-grass mixtures, there was no evidence for selecting certain texture features. This study was a first approach to using such information at all. This is emphasized in L. 307. In the future, a more systematic selection of the most important texture features may improve or simplify the biomass estimation models.

Vegetation index calculation

Please argue, why you have selected the mentioned 13 VI?

We used VIs which could be created with the available 4 bands and which were mentioned in literature for biomass estimation of vegetation and grassland. This was clarified in L. 322. 

2. Has the statistical analysis been performed appropriately and rigorously?

Yes.

3. Have the authors made all data underlying the findings in their manuscript fully available?

Not for the review.

4. Is the manuscript presented in an intelligible fashion and written in standard English?

Yes.

158: ……located in northern Hessen.

(L. 163) The authors disagree. In English the German state “Hessen” is called Hesse (though it sounds weird).

Reviewer #2: The manuscript is very well organised and the English style is good. The topic is interesting for the potential impact on operational precision farming. I have just few concerns:

1) It seems that the multispectral images were calibrated in reflectance using the irradiance sensor installed on the UAV. If this is true, then your images were calibrated in reflectance at platform level and not in ground reflectance. Since you are processing a time series of images this could be an important aspect to discuss. Also VI can be affected by the atmosphere, depending on their formulation. I think that this argument deserves a discussion;

The authors agree, that the atmosphere between sensor and ground can affect reflection. Nevertheless, flight height of 20-50 m was rather low and, thus, are the atmospheric effects on the reflectance. Additionally, it can be assumed that within the study area and given a very short flight time no fluctuation of the atmospheric conditions appeared. An atmospheric correction is rather used for satellite data and large areas, where atmospheric variation within the study area can be expected. Therefore, the authors would prefer to not discuss this in the manuscript.

2) In you analyses you are using the 13 VI as independent variables. However I would like to stress that these VI are usually correlated and for sure they are correlated with the original bands values. Please confirm or not that this is not a problem for the validity of your application, from statistical point of view.

To deal with multispectral data, which usually suffers from multicollinearity, machine learning algorithms are a solution. This method is commonly used in the field of remote sensing also with grasslands. For a better understanding, this is emphasized in L. 341.

Pag 17 - please check this sentence "The radiometric resolution was set to 0 to 255."

(L. 309) Changed to 16 bits.

---

## [Decision Letter · Decision Letter 1]

19 May 2020

PONE-D-19-34750R1

The potential of UAV-borne spectral and textural information for predicting aboveground biomass and N fixation in legume-grass mixtures

PLOS ONE

Dear Mrs. Grüner,

Thank you for submitting your manuscript to PLOS ONE. After careful consideration, we feel that it has merit but does not fully meet PLOS ONE’s publication criteria as it currently stands. Therefore, we invite you to submit a revised version of the manuscript that addresses the points raised during the review process.

We would appreciate receiving your revised manuscript by Jul 03 2020 11:59PM. To enhance the reproducibility of your results, we recommend that if applicable you deposit your laboratory protocols in protocols.io, where a protocol can be assigned its own identifier (DOI) such that it can be cited independently in the future. For instructions see: http://journals.plos.org/plosone/s/submission-guidelines#loc-laboratory-protocols

We look forward to receiving your revised manuscript.

Kind regards,

Jana Müllerová, Ph.D

Academic Editor

PLOS ONE

Additional Editor Comments (if provided):

Please add a short comment to the Discussion on the effects of atmosphere on your results

Reviewers' comments:

Reviewer's Responses to Questions

**Comments to the Author**

1. If the authors have adequately addressed your comments raised in a previous round of review and you feel that this manuscript is now acceptable for publication, you may indicate that here to bypass the “Comments to the Author” section, enter your conflict of interest statement in the “Confidential to Editor” section, and submit your "Accept" recommendation.

Reviewer #1: All comments have been addressed

Reviewer #2: All comments have been addressed

2. Is the manuscript technically sound, and do the data support the conclusions?

Reviewer #1: Yes

Reviewer #2: Yes

3. Has the statistical analysis been performed appropriately and rigorously? 

Reviewer #1: Yes

Reviewer #2: Yes

4. Have the authors made all data underlying the findings in their manuscript fully available?

Reviewer #1: No

Reviewer #2: (No Response)

5. Is the manuscript presented in an intelligible fashion and written in standard English?

Reviewer #1: Yes

Reviewer #2: Yes

6. Review Comments to the Author

Reviewer #1: I really appreciate the work the authors have been performed in order to critically reflect and significantly have improved the research presented. From my point of view all comments have been clearly addressed and I recommend this contribution for publication.

Reviewer #2: Dear Authors,

it is true that a flight height of 20-50 m and a limited short flight duration (and small area) can suggest to do not apply any atmospheric correction. However my concern was about the processing of a time series of images. From one image time acquisition to another the atmospheric influence could be different. And this can affect the VI values. Please just shortly discuss this topic and how this could affect your analysis.

7. PLOS authors have the option to publish the peer review history of their article (what does this mean?). If published, this will include your full peer review and any attached files.

Reviewer #1: No

Reviewer #2: No

---

## [Author Response · Author response to Decision Letter 1]

22 May 2020

Manuscript ID: PONE-D-19-34750R1 - Response letter

Dear Mrs. Müllerová,

Dear Reviewers,

thank you very much for the second review of our manuscript titled „ The potential of UAV-borne spectral and textural information for predicting aboveground biomass and N fixation in legume-grass mixtures”. We have carefully implemented the comments. 

Please find our specific answer to each reviewer’s comments below in bold letters.

Yours sincerely

Esther Grüner

Reviewers' comments:

Reviewer #1: I really appreciate the work the authors have been performed in order to critically reflect and significantly have improved the research presented. From my point of view all comments have been clearly addressed and I recommend this contribution for publication.

Thank you.

Reviewer #2: Dear Authors,

it is true that a flight height of 20-50 m and a limited short flight duration (and small area) can suggest to do not apply any atmospheric correction. However my concern was about the processing of a time series of images. From one image time acquisition to another the atmospheric influence could be different. And this can affect the VI values. Please just shortly discuss this topic and how this could affect your analysis.

We have carefully implemented this comment to L. 572-577 in the section “Discussion”.

---

## [Editor Report · Decision Letter 2]

2 Jun 2020

The potential of UAV-borne spectral and textural information for predicting aboveground biomass and N fixation in legume-grass mixtures

PONE-D-19-34750R2

Dear Dr. Grüner,

We’re pleased to inform you that your manuscript has been judged scientifically suitable for publication and will be formally accepted for publication once it meets all outstanding technical requirements.

Kind regards,

Jana Müllerová, Ph.D

Academic Editor

PLOS ONE
---

## [Editor Report · Acceptance letter]

15 Jun 2020

PONE-D-19-34750R2 

The potential of UAV-borne spectral and textural information for predicting aboveground biomass and N fixation in legume-grass mixtures 

Dear Dr. Grüner:

I'm pleased to inform you that your manuscript has been deemed suitable for publication in PLOS ONE. Congratulations! Your manuscript is now with our production department. 

Kind regards, 

on behalf of

Dr. Jana Müllerová 

Academic Editor

PLOS ONE